

# Annual cycle in scots pine's photosynthesis

Pertti Hari[1], Veli-Matti Kerminen[2], Liisa Kulmala[1], Markku Kulmala[2], Steffen Noe[3], Tuukka Petäjä[2], Anni Vanhatalo[1], Jaana Bäck[1]

[1] Department of Forest Sciences, University of Helsinki, Helsinki, P.O. Box 27, FI-00014 University of Helsinki, Finland

[2] Department of Physics, University of Helsinki, Helsinki, P.O. Box 64, FI-00014 University of Helsinki, Finland

[3] Department of Plant Physiology, Estonian University of Life Sciences, Tartu, Fr.R. Kreutzwaldi 1, EE-51014 Tartu, Estonia

*Correspondence to*: Pertti Hari (pertti.hari@helsinki.fi)

**Abstract.** Photosynthesis, i.e. the assimilation of atmospheric carbon to organic molecules with the help of solar energy, is a fundamental and well understood process. Here, we connect theoretically the fundamental concepts affecting C3 photosynthesis with the main environmental drivers (ambient temperature and solar light intensity), using six axioms based on physiological and physical knowledge and yield straightforward and simple mathematical equations. The light and carbon reactions in photosynthesis are based on the coherent operation of the photosynthetic machinery, which is formed of a complicated chain of enzymes, membrane pumps and pigments. A powerful biochemical regulation system has emerged in evolution to match photosynthesis with the annual cycle in solar light and temperature. The action of the biochemical regulation system generates the annual cycle of photosynthesis and emergent properties, the state of photosynthetic machinery, and the efficiency of photosynthesis. The state and the efficiency of the photosynthetic machinery is dynamically changing due to biosynthesis and decomposition of the molecules. The mathematical analysis of the system, defined by the very fundamental concepts and axioms, resulted in exact predictions of the behaviour of daily and annual patterns in photosynthesis. We tested the predictions with extensive field measurements of Scots pine photosynthesis in Northern Finland. Our theory gained strong corroboration for the theory in the rigorous test.



# 1 Introduction

The movement of the globe around the sun generates a conspicuous annual cycle in the solar radiation on the earth, and this cycle is especially strong at high latitudes. Ambient temperatures respond to the cycle in solar energy input and therefore a strong annual cycle exists also in temperature, although a bit delayed. These large variations in light and temperature are

greatly influencing the distribution of plant species, especially in the northern regions. As an example, Scots pines (*Pinus sylvestris* L.), while abundant all over the Europe, have adapted especially well also to the annual cycle of radiation and temperature in the northern climate, forming even the treeline in many regions (Juntunen et al. 2002).

As a consequence of the seasonal variation in light and temperature, many perennials including deciduous trees have a strong metabolic annual cycle, as they grow new leaves every spring that then become senescent in the autumn. Temperature

affects the timing of many phenological events i.e. bud burst and flowering (Hänninen and Kramer 2007, Hari and Häkkinen 1991, Linkosalo 2000, Sarvas 1972). However, the annual cycle is less clear in coniferous trees, although they also have a period of intensive new foliage growth in the spring, and a specific time frame when old needles are senescing in the fall.

The annual cycle of light and temperature is manifested in plant metabolism in many ways. Actively metabolizing cells are very sensitive to low temperatures, and, as a consequence, they need to inactivate many processes in order to avoid damage

during winter in a process called winter hardening (Hänninen 2016). This means that the metabolism of e.g. evergreen Scots pine needles also needs to follow a clear annual cycle. For example, when sufficiently hardened, pine needles tolerate temperatures well below –30 °C in winter, however in summer they are very sensitive to temperatures below –10 °C during summer (Sakai and Larcher 1987). The metabolism of photosynthesis recovers gradually from the winter hardened state during spring, and the ambient temperature has an important role in this recovery (Pelkonen and Hari 1980).

Biochemically, photosynthesis can be defined as a long chain of action of pigments, membrane pumps and enzymes, which use light as source for energy and atmospheric $CO_2$ as source for carbon (see e.g. (von Caemmerer and Farquhar 1981)). Changes in the concentrations and activities of this photosynthetic machinery generate the annual metabolic cycle in photosynthesis. The physiological basis of the annual cycle at the level of the synchronized action of pigments, membrane pumps and enzymes is poorly known, especially when it comes to the role of temperature in the synthesis, activation,

decomposition and deactivation of the machinery.

Sugars formed in photosynthesis are the source of energy for all cellular metabolic activity and raw material for growth. The length of the photosynthetically active period is a key factor determining the annual amounts of sugars formed in photosynthesis (Hari et al. 2013) and it plays a very important role in the metabolism and growth of vegetation. Thus, a theoretical understanding of the dynamics of the photosynthetic annual cycle is a key to understanding and explaining the

growth of the trees growing at high latitudes.



Physiological and biochemical research has provided useful knowledge of the photosynthetic reaction chains, and the details of this machinery at leaf, organ and tissue levels have been intensively explored over decades, mostly in controlled, laboratory conditions (Farquhar and von Caemmerer 1982, Farquhar et al. 1980, Kirschbaum et al. 1998, Laisk and Oja 1998). However, field measurements in mature trees are difficult to perform, and the results are not easy to interpret.

Therefore, the detailed physiological knowledge that has mostly been obtained from laboratory experiments needs to be translated into the ecological level to increase our understanding on the annual cycle of photosynthesis under field conditions. This was our motivation in developing a conceptual approach to the relationship of photosynthesis and the annual variations in light and temperature.

Physics was facing a similar situation in the seventeenth century as field studies on photosynthesis are encountering now.

There were plenty of single and scattered experiments and observations, but the unifying theory was missing. Isaac Newton presented an approach to construct theories in his book Principia Mathematica and unified the physical knowledge. He proceeded in four steps when developing theories, starting from the definition of concepts and followed by the introduction of axioms. The mathematical analysis of the behaviour of the system defined by the concepts and axioms dominated his theory development. Finally, he derived predictions and tested them. The new translation of Newton's famous book

*Principia Mathematica* (Newton 1999) clearly presents these four steps.

In our previous analysis of photosynthesis taking place during midsummer, we followed strictly Newton's example by introducing the concepts and axioms, by analysing the behaviour of the system defined by these concepts and axioms, and finally by deriving predictions and testing them (Hari et al. 2014). However, it was evident that our theory omits the annual cycle of metabolism and therefore it fails crucially to predict the photosynthesis in the transitional times such as spring and

autumn. The daily patterns of measured and predicted $CO_2$ exchange were quite similar, but the level of predicted photosynthesis was too low, especially in early spring and late autumn. We thus concluded that we have to introduce the annual cycle of metabolism into our theory. Our aim is to develop our theory of photosynthesis to cover the whole growing season and to explain and to predict the annual cycle of Scots pine photosynthesis in field conditions.

**2 Theory development**

The strong annual cycle in the solar light intensity and ambient temperature is characteristic for the growing area of Scots pine: for example in Finland, summers are quite mild, daily maximum temperatures being around 20 °C, whereas winters are rather cold with minimum temperature often below –20 °C. A regulation system has emerged in evolution to match the metabolism and cold tolerance with the annual cycle in the solar radiation and temperature.



The process of photosynthesis consists of a large number of steps that form the light and carbon reactions of photosynthesis. Each step is based on actions of a specific molecule, the most important being pigments (e.g. chlorophylls and carotenoids), transmembrane proteins and membrane pumps (e.g. ATPases), and Calvin cycle enzymes (e.g. ribulose-1,6-bisphophatase, Rubisco) (Taiz et al. 2015). A proper functioning of the reaction chain in photosynthesis requires that no single step is

blocking the chain of interlinked energy capture, membrane transport or synthesis of new compounds. The core of pigment complexes, as well as the membrane pumps and enzymes are all proteins that have a tendency to decay (Araujo et al. 2011, Hinkson and Elias 2011, Huffaker and Peterson 1974, Nelson et al. 2014). Proteins are nitrogen-rich macromolecules (many contain 15–16 w-% N (Nelson et al. 2014)) and they are costly to produce and maintain. Therefore, it is natural that plants need to be able to use the limited N reserves in an effective way. Since nitrogen has several competing usages in the

metabolism, maintaining excess proteins is a 'waste' of nitrogen. Synthesis and decomposition of active protein molecules balance the concentrations of active protein molecules in the photosynthetic chain. Evidently, maintaining the proper balance of these molecules is a crucial and demanding task for the metabolism of trees.

Large changes in the photosynthetic performance characterize the annual cycle of photosynthesis, generated by changes in the concentrations of pigments, membrane pumps and enzymes. Maintaining the proper concentrations of pigments,

membrane pumps and enzymes is taken care by a very powerful biological regulation system that has emerged in the evolution to match the cellular metabolism with the regular annual cycle in solar light and temperature (Ensminger et al. 2004a). This system synthetises, activates, decomposes and deactivates the critical pigments, membrane pumps and enzymes over time scales of days (Nelson et al. 2014), and it is an acclimation system, affecting the activation and deactivation of transcriptional modules responsive to light and temperature cues (e.g. (Cazzonelli and Pogson 2010, Toledo-

Ortiz et al. 2014)). The changes in the enzymes, membrane pumps and pigments, in turn, generate changes in the relationship between photosynthesis and light. This forms the metabolic basis for our theory of the dynamics of annual cycle of photosynthesis.

## 2.1 Definitions and axioms

We start our formulation with definitions as Newton did centuries ago. We utilise physiological and physical knowledge in the formulation of the axioms needed for the mathematical formulation.

**Definition 1.** We call the complex web of pigments, membrane pumps and enzymes forming the biochemical structure underlying photosynthesis as photosynthetic machinery.

Plants are able to change the concentrations of active components in the photosynthetic machinery.



**Definition 2.** Plants have a biochemical regulation system that synthetize, activate, decompose and deactivate the photosynthetic machinery.

The action of the biochemical regulation system generates the annual cycle in photosynthesis and maintains the balance between the different steps in the photosynthetic reaction chain. In this way, it generates a new property in the

photosynthetic machinery.

**Definition 3.** The action of the biochemical regulation system generates an emergent property, in the concentrations of active enzymes, membrane pumps and pigments, called the annual state of the photosynthetic machinery.

The state of photosynthetic machinery characterises the complex web of energy capture, biochemical reactions and membrane transport in photosynthesis with one single number. Next, we specify the action of biochemical regulation system

on photosynthetic machinery:

**Axiom 1.** Synthesis and activation, and decomposition and deactivation of enzymes, membrane pumps and pigments are changing the annual state of the photosynthetic machinery.

Further, we specify the relationship between environment and the synthesis by the biochemical regulation system.

**Axiom 2.** The synthesis and activation of enzymes, membrane pumps and pigments depend linearly on the temperature

above freezing point.

We clarify also the behaviour of decomposition and deactivation.

**Axiom 3.** The decomposition and deactivation of enzymes, membrane pumps and pigments depends linearly on the state.

Captured light energy may cause damage in chloroplasts in freezing temperatures, when availability of $CO_2$ is limited for the carbon reactions in photosynthesis. This is why the biochemical regulation system acts strongly to protect against damage.

**Axiom 4.** The accelerated decomposition and deactivation of enzymes, membrane pumps and pigments during cold and strong light depends linearly on the product of light and temperature below freezing point.

The concentrations of pigments, membrane pumps and enzymes affect the performance of photosynthesis.

**Definition 4.** The efficiency of photosynthetic reactions is the capacity of light and carbon reactions to synthesise sugars.

When we developed the theory of photosynthesis explaining the behaviour in midsummer (Hari et al. 2014), we introduced

an axiom stating that the product of saturating response to the photosynthetically active radiation and $CO_2$ concentration in the stomatal cavity determines the photosynthesis at a point in space and time. Here, we introduce the annual cycle of



photosynthesis into the axioms with the efficiency of photosynthetic carbon and light reactions and the efficiency photosynthetic reactions replace the parameter b in the Eq.(1) in Hari et al. 2014.

**Axiom 5.** The photosynthesis rate at a point in space and time depends on the product of two terms: i) the efficiency of photosynthetic light and carbon reactions, and ii) the product of $CO_2$ concentration in the stomatal cavity and the saturating response of the light reactions to the photosynthetically active radiation.

The state of the photosynthetic machinery determines the efficiency of photosynthetic light and carbon reactions, which leads to our final axiom:

**Axiom 6.** The efficiency of photosynthetic light and carbon reactions depends linearly on the state of the photosynthetic machinery.

**2.2. Mathematical analysis**

We introduce mathematical symbols to formulate exactly the axioms in a more exact and compact way. Let $S$ denote the state of the photosynthetic machinery, $f_1$ is the synthesis and activation, $f_2$ is the decomposition and deactivation of enzymes, membrane pumps and pigments, and is $f_3$ the accelerated decomposition and deactivation of enzymes, membrane pumps and pigments caused by light at low temperatures.

Axiom 2 states that synthesis and activation depends linearly on temperature ($T$) above the freezing point, which gives:

$$f_1(T) = \text{Max}\{0, a_1(T + T_f)\},$$
(1)

where $T_f$ is the freezing temperature of needles and $a_i$ is a parameter.

According to axiom 3, the decomposition and deactivation of photosynthetic machinery depends linearly on the state of photosynthetic machinery, $S$:

$$f_2(S) = a_2 S.$$
(2)

Accelerated decomposition and deactivation takes place to protect the photosynthetic machinery against damage when freezing temperatures hinders the carbon assimilation reactions of photosynthesis (Axiom 4):

$$f_3(I, T) = a_3 \text{Max}\{(T_f - T) I, 0\},$$
(3)

where $I$ is the intensity of photosynthetically active radiation.

The synthesis, activation, decomposition and deactivation change the state of the photosynthetic machinery, as follows:



$$\frac{dS}{dt} = f_1 - f_2 - f_3$$

(4)

Combining Equations (1)-(4), we obtain:

$$\frac{dS}{dt} = \text{Max}\{0, a_1(T + T_f)\} - a_2 S - a_3 \text{Max}\{(T_f - T) I, 0\}$$

(5)

Equation (5) defines the state of the photosynthetic machinery at any moment $t$ when temperature and solar radiation records

are available.

The photosynthesis rate, $p$, is obtained from the axiom 5, as follows:

$$p = E f(I) C_S,$$

(6)

where $C_s$ is the $CO_2$ concentration in the stomatal cavity, $f(I)$ is the saturating response of the photosynthesis rate to the

photosynthetically active radiation (see Hari et al 2014), and $E$ is the efficiency of photosynthetic carbon and light reactions

which, according to the axiom 6, it is as follows:

$$E = a_4 S$$

(7)

When we developed the theory of photosynthesis in midsummer (Hari et al. 2014), we introduced an axiom stating that the

product of saturating response to the photosynthetically active radiation and $CO_2$ concentration in the stomatal cavity

determines the photosynthesis at a point in space and time (A1 in Hari et al. 2014). Here, we replaced the axiom A1 with the

new axiom 5 that is quite similar with the previous one. The changing efficiency of photosynthetic light and carbon reactions

is the novel aspect in the axiom 6. When we quantified with mathematical notations the previous axiom, we introduced a

parameter $b$ (Eq. 1 in Hari et al. 2014). Equation (6) is very similar with the previous Eq. (1) in Hari et al. (2014); the only

difference is that the efficiency parameter $b$ is replaced with $E$, the state variable efficiency of photosynthetic carbon and

light reactions. We obtain the solution of the optimisation problem in the same way as in the analysis of photosynthesis ($p$)

during midsummer, as follows:

$$p(I, E) = \frac{(u_{opt}\, g_{max} C_a + r) a_4\, S\, f(I)}{u_{opt}\, g_{max} + a_4\, S\, f(I)},$$

(8)

where $g_{max}$ is stomatal conductance when stomata are open, $C_a$ $CO_2$ concentration in the ambient air, $r$ is the rate of

respiration and $u_{opt}$ is so-called seasonal modulated degree of optimal stomatal control given by

$$u_{opt} = \begin{cases} 0, & if \ u \leq 0 \\ u, & if \ 0 < u \leq 1 \\ 1, & if \ u > 1 \end{cases}$$

(9)



$$u = \left( \sqrt{\frac{c_a - r/(a_4\, S\, f())}{\lambda\, a\, (e_s - e_a)}} \right) \frac{a_4\, S\, f()}{g_{max}}$$

(10)

In the Eq. (10), $\lambda$ is a cost of transpiration i.e. a measure of water-use efficiency.

To summarize, Eqs. (5), (7) - (10) predict the density of photosynthetic rate when we know the ambient temperature and solar radiation history, density of photosynthetically active solar radiation, and concentrations of water vapour and $CO_2$ in

the air. This prediction is clearly a dynamic version of the formulation by Hari et al (2014). The changing state of the photosynthetic machinery (i.e. enzymes, membrane pumps and pigments) determines the efficiency of light and carbon reactions, introducing the annual cycle of metabolism into the prediction. Thus, the relationship between light and photosynthesis changes smoothly during the seasons.

### 2.3. Parameter estimation

We tested the new theoretical prediction with field chamber measurements at Scots pine trees in Lapland, Värriö Subarctic Research Station (67°46'N, 29°35'E, 400 m a.s.l). We measured the $CO_2$ exchange of pine shoots with four branch chambers throughout the year in 2011-2014 (Hari et al. 2014). Despite the constant supervision, maintenance and malfunction of the measuring system generated some gaps in the data. To obtain maximal data coverage per year, we selected those chambers that measured over the whole year without long maintenance and malfunction periods.

There are four parameters in the Eqs. (5), (7) - (10) that describe the annual cycle of photosynthesis ($a_1$, $a_2$, $a_3$ and $a_4$). The freezing temperatures and sunny weather are quite rare events at our measuring station occurring only in early spring and very late in autumn. As a result, the parameter $a_3$ in the Eq. (3) has a minor role in the predictions and its estimation is based on very scarce data. Local minima of the residual sum of squares disturbed simultaneous estimation of the parameters $a_1$, $a_2$, and $a_4$. Therefore, we used a graphical method in the estimation and we fixed the value of the parameter $a_2$. Then we

automatized the estimation of $a_1$ and $a_4$. The estimation resulted into the following values: $a_1 = 10$, $a_2 = 0.065$, $a_3 = 2$ and $T_f = -3$. The values of the parameter $a_4$ are year and chamber specific.

### 3 Results

We predicted the state of the photosynthetic machinery i.e. the annual state of enzymes, membrane pumps and pigments with the Eq. (5) using the measured temperature and light intensity before the moment in consideration. The predicted annual

patterns of the state of the photosynthetic machinery were quite similar between the different years (Fig. 1). There was,



however, some weather-driven variation. For example, the very warm August in 2014 generated the large peak in late summer.

The changes in the relationship between light and photosynthesis is characteristic to our theory. Figure 2 depicts the daily patterns of the measured and predicted leaf $CO_2$ exchange early in the spring (A) and at midsummer (B). The measured and predicted daily patterns generated by the variation in light were very similar to each other, although the level of photosynthesis increased considerably from spring to summer. Our theory predicted the level of this increase during the summer successfully.

Days of intermittent cloudiness dominate our northern climate in the summer (Hari et al. 2014), giving rise to very strong within-day variations in the light levels reaching the canopy. Our theory predicted strong variation in photosynthesis during days of intermittent cloudiness, yet the measured leaf $CO_2$ exchange seemed to be very similar with the predicted one (Fig. 3A).

Heavy clouds tend to cover the sky during rainy days strongly reducing the light intensity. Our theory predicts strongly reduced photosynthesis during dark rainy days. Again, the measured and predicted leaf $CO_2$ exchange were very close to each other when thick clouds covered the sky (Fig. 3B).

Our theory predicts clear effect of partial closure of stomata on sunny days when the temperature strongly increases during the day. This type of days are, however, rather rare events at our northern measuring site. Nevertheless, the measurements of leaf $CO_2$ exchange showed a similar pattern with our prediction on such days (Fig. 3C).

We have continuous measurements for four summers, consisting of more than 30 000 data points during each summer. The predictions of leaf $CO_2$ exchange of a shoot were very close to the measured pattern, without exception. Also, the relationships between measured and predicted leaf $CO_2$ exchange indicated close correlations between measurements and predictions (Fig. 4). The predictions explained about 95 % of the variance of the measured values.

The residuals, i.e. the difference between measured and predicted leaf $CO_2$ exchange revealed only slight systematic behaviour (Fig. 5) indicating that the theory was a quite adequate description of the regularities in the photosynthesis of northern Scots pine.

**4 Discussion**





Scots pine has a broad distribution range all over Europe, and the local populations have adapted to the regular annual cycle in solar radiation and in temperature. The needle metabolism has also a clear annual cycle that alternates between the cold tolerance and very low metabolic activity during winter and strong metabolism and cold vulnerability in summer. The annual cycle is particularly strong in photosynthesis (Ensminger et al. 2004b, Kolari et al. 2014, Öquist and Huner 2003, Pelkonen and Hari 1980).

We have worked decades with the annual cycle of vegetation from the analysis of daily shoot elongation (Hari and Leikola 1972, Hari et al. 1977), bud burst of trees (Hari and Häkkinen 1991) and photosynthesis (Pelkonen and Hari 1980). Our approach has been dynamic modelling without clear connection to the physiological background, although we were looking for the metabolic explanations. The strong connection to the light and carbon reactions and their basis on enzymes, membrane pumps and pigments is the novel feature of our theory of the annual cycle of photosynthesis. It provides sound physiological background to our concepts and axioms. We utilised strongly physiological knowledge in the development of our theory. Previously the focus has been in the mathematical formulation of the ideas whereas the physiological background has been quite unclear. The predictions of our novel theory are close to those obtained previously (Mäkelä et al. 2004) although the fit of the predictions with measurements has improved considerably.

The light and carbon reactions of photosynthesis are down regulated in autumn in order to protect the sensitive machinery against low temperatures, and up regulated again in spring. This seasonality has been closely connected to variations in ambient temperatures (Mäkelä et al. 2004, Pelkonen and Hari 1980) and photoperiod or light intensity changes (Ensminger et al. 2004a, Porcar-Castell et al. 2008). A delayed effect of temperature on photosynthesis recovery in spring is introduced (Mäkelä et al. 2004, Pelkonen and Hari 1980) and tested with field measurements (Kolari et al. 2009).

The Newtonian approach provided a sound backbone to collect physiological knowledge for the development of our theory of annual cycle of photosynthesis. The definitions of concepts determine the most important features in the theory and the axioms the critical relationships between the concepts. Applying mathematical analysis and simulations of the behaviour of the system, as defined by the concepts and axioms, proved to be an efficient tool to analyse the consequences in photosynthesis and to derive predictions.

We defined new concepts, the biochemical regulation system and the state of photosynthetic machinery (enzymes, membrane pumps and pigments) that played very important role in the argumentation. The physiological basis of the new concept is clear, since large number of steps form the light and carbon reactions of photosynthesis. Each step is based on specific pigment, membrane pump or enzyme. In efficient metabolic chain, the steps have to be in balance with each other. The biochemical regulation system, emerged in evolution, generates balance between the steps in photosynthesis whereas its action generate the state of the photosynthetic machinery. The state of the photosynthetic machinery determines the



changing efficiency of the light and carbon reactions in photosynthesis. In this way, the action of the biochemical regulation system generates the annual metabolic cycle in photosynthesis and the synchrony with the strong annual cycle in radiation and temperature.

The axioms clarify the action of the biochemical regulation system in synthesis and decomposition of photosynthetic

machinery. The physiological basis of the actions is clear. Metabolic reactions take place faster at elevated temperatures than in low ones. Thus synthesis is temperature dependent (Axiom 2). The enzymes, membrane pumps and pigments are non-stable compounds as introduced in the axiom 3.

The increasing temperatures in the spring accelerate the synthesis and activation of photosynthetic machinery resulting in increasing photosynthesis. The combination of sunny and cold mornings is frequent at our field station then the accelerated

decomposition and deactivation strongly decrease photosynthesis. When the spring proceeds, air temperature rises and the synthesis and activation increase the state of the photosynthetic machinery resulting in enhanced photosynthesis.

The enzymes, membrane pumps and pigments are non-stable compounds and consequently, their decomposition and deactivation increases during summer resulting into quite stable state of the photosynthetic machinery. When the temperature starts to decrease according the annual cycle, the synthesis declines decreasing the pool of these non-stable

compounds resulting in a reduction in the light response of photosynthesis. In this way the biochemical regulation system generates the annual metabolic cycle in photosynthesis that is in delayed synchrony with the annual cycle of radiation and temperature.

Our theory predicts slow recovery in the spring, quite steady maximum in the summer and slow decline in the autumn to be characteristic for the annual cycle of photosynthesis due the synthesis, activation, decomposition and deactivation of

photosynthetic machinery. The observed annual patterns of photosynthesis are in an agreement with the above theoretical prediction.

The diurnal cycle in radiation and temperature is clear in summer time and missing during the polar night at our research site. However, we can omit the polar night in photosynthetic studies due to darkness and low temperatures. Our theory predicts that (i) photosynthesis during a day follows the saturating response to light, since the changes in the concentrations

of enzymes, membrane pumps and pigments are so slow that the changes do not affect the behaviour of photosynthesis during a day and (ii) the action of stomata slows down photosynthesis during most sunny days. Our field measurements are in agreement with this prediction.

Our theory has passed successfully the above qualitative tests. However, quantitative tests are more severe and they can provide stronger corroboration for the theory. We tested our theory with field measurements over four years including over

130 000 measurements of $CO_2$ exchange, PAR, temperature, atmospheric $CO_2$ and water vapour concentration. Our theory



predicted the annual and daily patterns of photosynthesis explaining about 95 % of the variance in the measured $CO_2$ exchange whereas residuals did not show any clear systematic behaviour. Thus our theory passed successfully the sever tests also in quantitative terms.

Short field campaigns and statistical analysis of the obtained data dominates photosynthetic research in the field. The short and fragmentary measurement series hinder the studies of the annual cycle of photosynthesis. The smoothly changing relationship between light and photosynthesis is missing in most statistical analysis of field measurements. The slow changes in the studied relationship are problematic for the statistical analysis of field data and probably explain why there is not any comparable ecological theory of annual photosynthesis.

In conclusion: Scots pine has adapted to the regular annual cycles in light and temperature and the effective biochemical regulation system of photosynthetic machinery has emerged in the evolution. The action of the biochemical regulation system generates the delayed annual cycle in photosynthesis by synthetizing, activating, decomposing and deactivating enzymes, membrane pumps and pigments. The linear relationship between synthesis and activation on temperature above the freezing point synchronises the metabolic and light cycles with each other. Prevailing light and the annual metabolic cycle determines photosynthesis, although the action by the stomata modifies the photosynthetic response. Our extensive field measurements corroborate the above conclusion.

## Data availability

All measurements at SMEARI including also the shoot chamber measurements are available from
https://avaa.tdata.fi/web/smart/smear/download. The code is available in Mathematica and can be accessed via the
corresponding author (pertti.hari@helsinki.fi).

## Acknowledgements

Acknowledgements: The research was funded by the Academy of Finland Center of Excellence programme (grants no 1118615 and 272041) Academy of Finland (277623), Nordic Center of Excellence program (CRAICC – Cryosphere-Atmosphere Interactions in a Changing Arctic Climate), ERC Advanced Grant No. 227463 ATMNUCLE, and the Nordforsk CRAICC-PEEX Nordic-Russian Cooperation programme. We thank Pasi Kolari and the staff at SMARI for the maintenance of the instruments and data.



**Competing interests**

The authors declare that they have no conflict of interest.

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



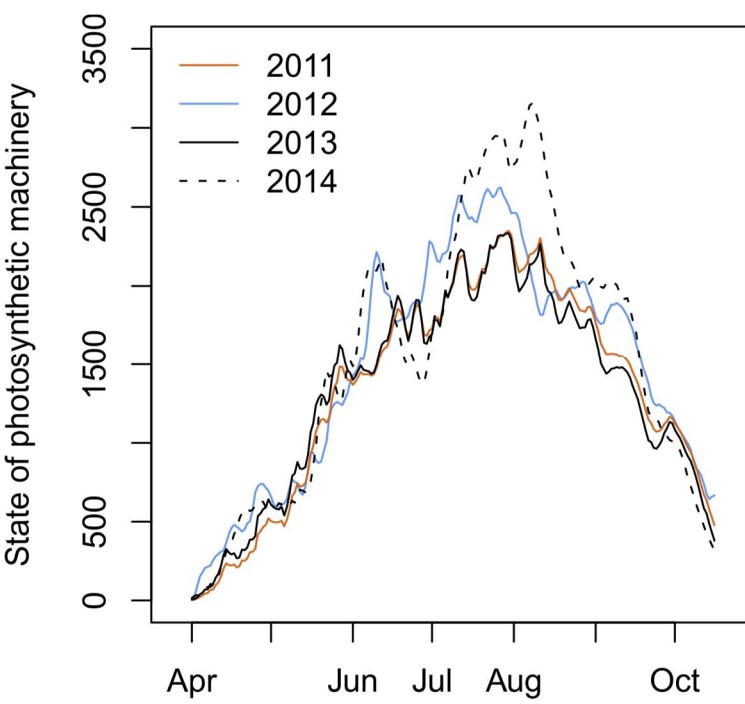

**Figure 1.** **The annual pattern of the state of the photosynthetic machinery (S, arbitrary units) during the years 2011–2014 in Finnish Lapland, 68$^{\circ}$N.**



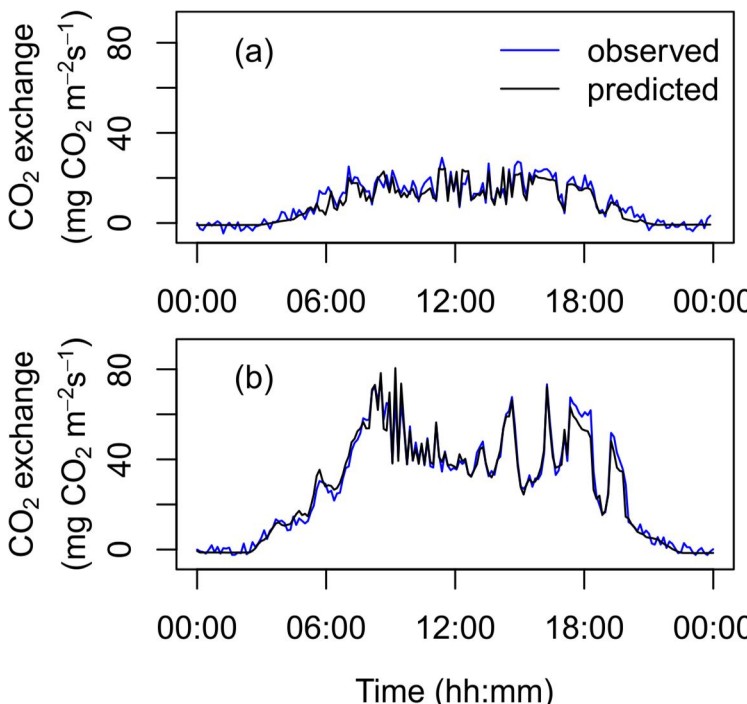

**Figure 2. Measured and predicted leaf CO₂ exchange during two days: a) early in the spring (May 8) and b) in midsummer (July 18) in Finnish Lapland, 68N.**





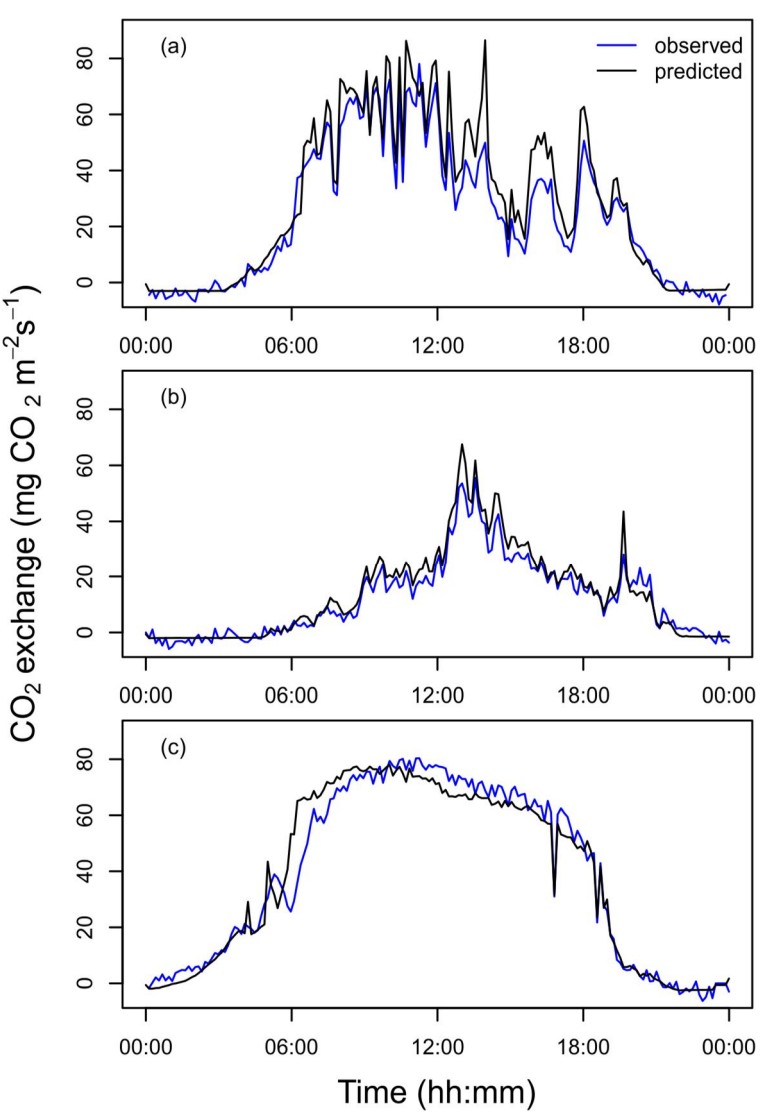

**Figure 3.** **Measured and predicted leaf $CO_2$ exchange (a) during a day of intermittent cloudiness (August 5), (b) during a cloudy day (July 22), and (c) during a sunny day when the stomata close partially (July 7) in Finnish Lapland, 68N.**



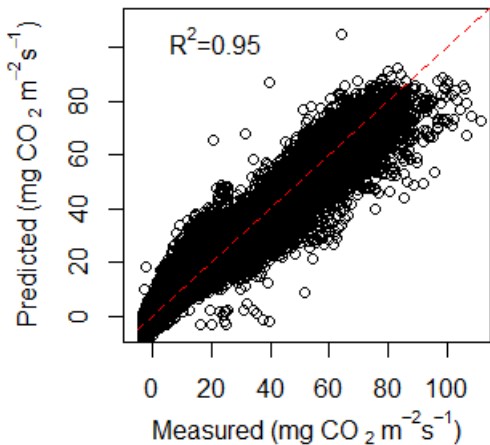

**Figure 4. Relationship between measured and predicted leaf $CO_2$ exchange in Finnish Lapland, 68N in the year 2013. The dashed line represents 1:1 line.**





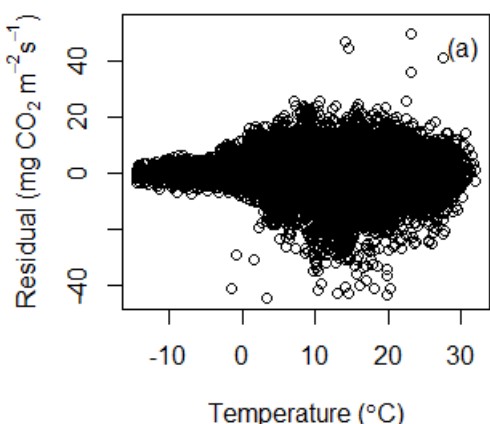

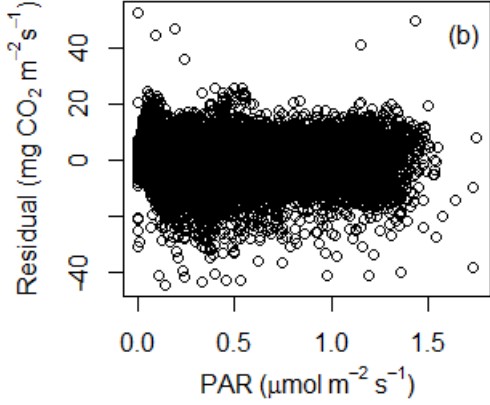

**Figure 5.** The residuals as function of temperature and PAR in the year 2013 in Finnish Lapland, 68N.