# Peer review of "Annual cycle in Scots pine's photosynthesis"

_Atmospheric Chemistry and Physics, 2017_

## Referee Comment (RC1) · K. Pilegaard (Referee) · 14 Aug 2017

**1   General comments**

A mathematical model describing the annual course of photosynthesis in Scots pine was constructed from fundamental concepts and axioms describing the variation in photosynthesis with basic environmental drivers such as ambient temperature and solar light intensity. The mathematical model was tested against a multi year dataset from Northern Finland, which resulted in exact predictions of the daily and annual cycle in photosynthesis.

The theoretical framework is clearly described and the resulting equations seems quite meaningful. I miss a discussion of the meaning of the "constants" $(a_1...a_5)$. The estimation of the constants from the tuning of the model to the field data is not well described,

and it seems there were quite some challenges to this.

The "test" using the field data was not strictly independent, since the dataset was used to estimate the parameters of the model. Has any attempt been made to try the model on field data from other Scots pine stands, and would this result in other values of the parameters? And what determines the exact value of the parameters? Since the model is based on fundamental relationships between photosynthesis and light and temperature, a discussion of its universality would be interesting to include in the paper.

Overall, I find the paper very interesting and well argued. I think the paper could be approved and increase interest if the points mentioned above and in the specific comments are taken into consideration.

**2   Specific comments**

Title: Change "scots pine's" to "Scots pine's".

Abstract:
p.1, l.20: "Our theory gained strong corroboration for the theory ...": Not immediately meaningful; please re-formulate.

p.2, l.17-18: Delete one of the two instances of the word "summer".

p.3, l.6: Replace "on" with "of" (i.e. "of the annual cycle").

p.4, l.7-10: Considering the prominent role of nitrogen, I wonder why nitrogen is not mentioned directly in the axioms such as light and temperature. Is this because nitrogen is only considered to be internally circulated in the system?

p.5, l.7: Shouldn't it rather be "the **seasonal** state of the photosynthetic machinery"?

p.6, l.13: "is $f_3$" should be "$f_3$ is".

p.7, l.14: A more readable statement would be. "When we quantified the previous axiom with mathematical notations...".

p.8, l.15-21: The procedure for parameter estimation needs some more explanation. What is the exact "graphical method" used? Why was $a_2$ fixed and how was the value chosen. Exactly which of the measured values were used?

p.10, l.26-27: The sentence starting with: "The physiological bases ..." is unclear; is something missing?

p.10, l.28: Change to: "In **an** efficient metabolic chain".

p.10, l.29: Change to: "... the steps in **the** photosynthesis ...".

p.11, l.6: Change to "... that **at** low ones."

p.11, l.13: Change to: "... into **a** quite stable state ...".

p.11, l.14: Change to: "... according **to** the annual cycle ...".

p.11, l. 19: Change to: "... deactivation of **the** photosynthetic machinery.".

p.12, l. 2: It should probably read "severe".

p.12, l.24: Change to "SMEARI".

---

## Referee Comment (RC2) · K. Jõgiste (Referee) · 3 Sep 2017

The work presents substantial ideas about autotrophic production processes of the forest ecosystem. The testing of the theory has the central position in the scientific analysis. Basic assumptions in the model are presented in a strict order to capture essential logical behavior of the system.

Introduction: the idea about the modelling is presented! Page 3, line 5 – 6: What do we mean as an "ecological level" here? The modelling attempt based on the physiological data can aim the different level starting from one single organism stretching to landscapes and global ecosphere?

Theory development! Page 3, line 27: How the Finnish summers are supposed to be mild? The geographical extent of the country is very wide.

[Figure]

The evolutionary dynamics of life processes is highly varying: the idiosyncratic response of an organism, species or population to environmental conditions contains many possible solutions. I would like to have a more detailed comment on the limits of physiological reaction to annual cycle in light and temperature variation including extreme cases (page 4, line 16)? BTW: the population variation has been mentioned in the discussion part: page 10, line1. Results of the work discuss the variation at different levels: would it be useful attempt to describe variation with known and unknown source separately?

The methodology presented in the form of definitions and axioms is a brilliant idea. The wording and structure of the definitions and axioms can be improved in several cases. Definition 3 introduces the "emergent property": how this properties are organized (hierarchy, spatial or temporal generalization)?

I am a little confused by use of term "linear" (page 6)? What do we mean here: the linear function?

Results: One can judge the match between observed and predicted photosynthesis dependence on the cloudiness to be rather good. Why the highest overestimation happens in the afternoon with intermittent cloudiness (Figure 3A)?

Does the data from Värriö Subarctic Research Station include extreme cases or disturbance events: e.g. low temperature during the vegetation period or extreme droughts?

What are the actions mentioned in the discussion (page 11, line 4)? Semantically action refers to purposeful and systematic interplay between components of the system! Although the action (or operation) of the system can be interpreted as evolutionary developed property of a living organism, the biochemical mechanism (enzymes, pigments, membrane pumps) as such lacks the purpose oriented action?

In conclusion: the presented modelling is only a minor part of the research conducted during many years. The wider and more profound presentation of the study can be

found in other printed sources. Material presented with current manuscript is an elegant demonstration of powerful methodological tools to create better comprehending of complex nature of living world. I do recommend to accept the paper with some modifications.

Minor comments I suggest some improvements to the abstract: the repetition of "theory" in concluding sentence should be avoided.

Number of measurements: 30000 datapoints during a summer (page 9, line 18): is it connected to total record 130000 (page 11, line 30)?

Acronyms at the Acknowledgement part are understandable only for very few specialist: nevertheless the Google can provide more or less correct hints. Still, what is SMARI?

---

## Author Response (AR2)

**A point to point response to Kim Pilegaard (Referee)**

**1 General comments**

A mathematical model describing the annual course of photosynthesis in Scots pine was constructed from fundamental concepts and axioms describing the variation in photosynthesis with basic environmental drivers such as ambient temperature and solar light intensity. The mathematical model was tested against a multi year dataset from Northern Finland, which resulted in exact predictions of the daily and annual cycle in photosynthesis.

The theoretical framework is clearly described and the resulting equations seems quite meaningful. I miss a discussion of the meaning of the "constants" (a1...a5). The estimation of the constants from the tuning of the model to the field data is not well described, and it seems there were quite some challenges to this.

We added a short discussion on the meaning of the parameters in the revised manuscript as well as improved the methods section concerning the parameter estimation. We introduce these changes in the specific comments later.

The "test" using the field data was not strictly independent, since the dataset was used to estimate the parameters of the model. Has any attempt been made to try the model on field data from other Scots pine stands, and would this result in other values of the parameters?

This is a good idea and we have already analysed whole ecosystem scale fluxes (GPP) of other Scots pine sites with the same theory (Hari et al 2017 ACPD at https://www.atmos-chem-phys-discuss.net/acp-2017-533/) and with good results. In addition, we have rather similar shoot scale measurements from SMEAR II station in southern Finland but the measuring arrangements there differ from those at SMEAR I from where we catch undisturbed and more frequent branch chamber data. Thus, the evaluation of the model performance at SMEAR II would be different from that introduced in this study. In addition, we think that the results of this study are already interesting due to the far-north, harsh location of SMEAR I.

**And what determines the exact value of the parameters?**

We shortly discuss these issues in the revised manuscript but for some parameters such as  $a_1$  and  $a_2$  describing the synthesis and decomposition rate of the components in the photosynthetic machinery, such discussion would be too speculative at the moment and would require more experiments. However, the origin of  $a_3$ ,  $a_4$  and  $T_f$  are discussed now in the revised manuscript.

**Since the model is based on fundamental relationships between photosynthesis and light and temperature, a discussion of its universality would be interesting to include in the paper.**

Indeed, the model attempts to use a fundamental and very basic relationship between environmental conditions and branch carbon uptake. We have made another study where this branch scale model is used for predicting ecosystem scale fluxes in several Scots pine forests in different ecoclimatic regions (Hari et al 2017 ACPD). The model works well even with very different stands and can account for significant part of variation in CO2 fluxes in these sites. We added a short mention to this in the Discussion. Overall, I find the paper very interesting and well argued. I think the paper could be approved and increase interest if the points mentioned above and in the specific comments are taken into consideration.

2 Specific comments

Title: Change "scots pine's" to "Scots pine's".

**Corrected.**

Abstract:

p.1, I.20: "Our theory gained strong corroboration for the theory ...": Not immediately meaningful; please re-formulate.

We formulated it into the revised manuscript as "Our theory gained strong support in the rigorous test".

p.2, I.17-18: Delete one of the two instances of the word "summer".

Corrected.

p.3, I.6: Replace "on" with "of" (i.e. "of the annual cycle").

Corrected.

p.4, I.7-10: Considering the prominent role of nitrogen, I wonder why nitrogen is not mentioned directly in the axioms such as light and temperature. Is this because nitrogen is only considered to be internally circulated in the system?

The theory explains the daily and seasonal cycle of photosynthesis in an individual branch, and we assume, that the availability of nitrogen does not change these seasonal processes considerably within the scale we are using in our analyses. We have clarified the scale in the abstract and throughout the manuscript. It is known that nitrogen content of leaves is connected to the availability of nitrogen (fertility) of the stand and leaves with lower nitrogen content do have lower rate of photosynthesis. Thus, the nitrogen would steadily affect the overall level of photosynthesis and is linked to the parameter  $a_4$ . The reason we discuss the nitrogen here is as you suggest; we wanted to stress the role of internal nitrogen circulation within the branch in the building up of new protein rich compounds necessary for photosynthetic machinery and transport of the photosynthates.

p.5, I.7: Shouldn't it rather be "the seasonal state of the photosynthetic machinery"?

Here we had a mistake as well as in the following axiom 1. Those should be just "the state of the photosynthetic machinery" to be consistent in the analysis. These are corrected in the revised manuscript.

p.6, I.13: "is f3" should be "f3 is".

Corrected.

p.7, I.14: A more readable statement would be. "When we quantified the previous axiom with mathematical notations...".

**Corrected as suggested.**

p.8, I.15-21: The procedure for parameter estimation needs some more explanation. What is the exact "graphical method" used? Why was a2 fixed and how was the value chosen. Exactly which of the measured values were used?

We base our estimation on the minimization of the residual sum of squares. The residual sum of squares has several local minima and they hamper the estimation. We find easily the minima with numeric methods but the obtained parameter values vary greatly from one data set to another. Evidently, the local minima disturb the estimation. We developed estimation method that results in reasonable parameter values in all data sets available.

There are three parameter values to be estimated, when we fit our model with observed fluxes. We proceed step-wise, first we fix the value of a parameter. Thereafter we estimate the values of non-fixed parameters with standard numeric methods. We replace the value of the fixed parameter with the one obtained in the estimation. We select another parameter, fix its value with that one obtained in the previous round of estimation and estimate the other two parameters again. We continue the process of fixing estimating and replacing for several rounds until we get reasonable fit. In this way, we find the smallest one from a large number of local minima.

The estimation of the parameter values is quite problematic, since the behaviour of the residual sum of squares is very irregular and there are numerous local minima, which confuse the normal estimation with numeric methods. We therefore developed a method that selected smallest one from a large number of residual sums of squares. This method resulted quite stable solution of the minimization.

In the revised manuscript, we have improved the paragraph describing of the parameter estimation (the latter one in subchapter 2.3). In the revised text we do not use the questioned term "graphical method" since it is already described more openly and with more descriptive words. In addition, we re-wrote the estimation on  $T_f$  since it is actually an estimate obtained from a colleague and not really estimated in this study.

The needed measurements in the estimation are now stated in detail in the revised manuscript.

p.10, I.26-27: The sentence starting with: "The physiological bases ..." is unclear; is something missing?

We decided to drop the whole sentence and include main idea to the end of the previous one.

The old version: 'We defined new concepts, the biochemical regulation system and the state of photosynthetic machinery (enzymes, membrane pumps and pigments) that played very important role in the argumentation. The physiological basis of the new concept is clear, since large number of steps form the light and carbon reactions of photosynthesis.'

Revised version: 'We defined new concepts, the biochemical regulation system and the state of photosynthetic machinery (enzymes, membrane pumps and pigments) that played very important role in the argumentation and are justified from the basic physiological understanding of the photosynthetic processes.'

p.10, I.28: Change to: "In an efficient metabolic chain".

Corrected

p.10, I.29: Change to: "... the steps in the photosynthesis ...". Corrected.

p.11, I.6: Change to "... that at low ones."

Changed to "than at low ones"

p.11, l.13: Change to: "... into a quite stable state ...".

Corrected.

p.11, I.14: Change to: "... according to the annual cycle ...".

Corrected.

p.11, I. 19: Change to: "... deactivation of the photosynthetic machinery.". Corrected.

p.12, I. 2: It should probably read "severe".

You are right. We corrected it.

p.12, I.24: Change to "SMEARI".

Corrected.

**A point to point response to K. Jõgiste (Referee)**

The work presents substantial ideas about autotrophic production processes of the forest ecosystem. The testing of the theory has the central position in the scientific analysis. Basic assumptions in the model are presented in a strict order to capture essential logical behavior of the system.

Introduction: the idea about the modelling is presented! Page 3, line 5 - 6: What do we mean as an "ecological level" here? The modelling attempt based on the physiological data can aim the different level starting from one single organism stretching to landscapes and global ecosphere?

We meant trees in their natural environment with the ecological level. Since it was so vague term, we revised it to "*field conditions i.e. into trees living in their natural environment*" in the revised manuscript.

**Theory development! Page 3, line 27: How the Finnish summers are supposed to be mild? The geographical extent of the country is very wide.**

We agree, there is a difference in the temperature between southern Finland and Northern Finland, especially in degree days i.e. in the length of the warm season. However, except for the very southernmost coast, the whole country belongs to subarctic climate type according to the Köppen-Geiger climate classification that defines summer in this region to be mild. In addition, the difference in the mean maximum temperature in the summer is not that great. Thus we would still like to state that the summers are mild. However, we have re-phrased the section and included a references in the revised manuscript. Now, it states " for example Finland has mostly a subarctic climate according to Köppen-Geiger climate classification (Peel et al. 2007) meaning that summers are quite mild, daily maximum temperatures...".

The evolutionary dynamics of life processes is highly varying: the idiosyncratic response of an organism, species or population to environmental conditions contains many possible solutions. I would like to have a more detailed comment on the limits of physiological reaction to annual cycle in light and temperature variation including extreme cases (page 4, line 16)? BTW: the population variation has been mentioned in the discussion part: page 10, line1. Results of the work discuss the variation at different levels: would it be useful attempt to describe variation with known and unknown source separately?

The acclimation responses specifically discussed here are related to the seasonal dynamics of photosynthetic machinery, adapted to the harsh climate, and we show that they are providing resilience for the systems also during extreme conditions in the stressful winter-to-spring transition period. This has been clarified in the revised ms. However, we are not discussing the potential of these systems to provide protection in other times or for extreme events which last for longer periods, e.g. during summer droughts, although to certain extent these mechanisms also operate during the growing season. We consider this aspect to be out of the scope of this particular paper, although it is an interesting topic in itself. We are actually preparing an independent manuscript on the topic (Matkala et al, under preparation).

**The methodology presented in the form of definitions and axioms is a brilliant idea. The wording and structure of the definitions and axioms can be improved in several cases.**

According to your notice, we have reorganized the wording in the definitions 1 and 3 as well as in the axioms 1-4. In practise, we have 1) changed the word order to be more easy (for example '*We call ..... as the photosynthetic machinery*' was in the revised manuscript changed into '*The photosynthetic machinery is...*'. In addition, we tried to avoid the repetition of the phrase '*pigments, membrane pumps and enzymes*' by using '*the photosynthetic machinery*' that is already defined in the Definition 1. These changes clearly improved the readability of the axioms and definitions.

**Definition 3 introduces the "emergent property": how this properties are organized (hierarchy, spatial or temporal generalization)?**

This is an interesting aspect but after a consideration, we decided that we will keep the definition rather short and clear. However, the new wording of the definition introduces the hierarchy of the properties more clearly than the old one.

Old: 'The action of the biochemical regulation system generates an emergent property, in the concentrations of active enzymes, membrane pumps and pigments, called the annual state of the photosynthetic machinery.'

Revised: 'The state of the photosynthetic machinery is the emergent property created by the actions of the biochemical regulation system controlling the concentrations of active enzymes, membrane pumps and pigments.'

I am a little confused by use of term "linear" (page 6)? What do we mean here: the linear function?

We mean that the relationship between efficiency of photosynthetic light and carbon reactions is linear. We clarified this in the revised manuscript above the equations 1 and 2.

Results: One can judge the match between observed and predicted photosynthesis dependence on the cloudiness to be rather good. Why the highest overestimation happens in the afternoon with intermittent cloudiness (Figure 3A)?

True, interesting remark! However, we thought that the overestimation is so small that it is most probably generated by normal random variation and did not discuss on it in the manuscript.

**Does the data from Värriö Subarctic Research Station include extreme cases or disturbance events: e.g. low temperature during the vegetation period or extreme droughts?**

We have experienced a prolonged season with low soil moisture in 2013 which was exceptional in the area. Usually the area is very humid since precipitation exceeds evapotranspiration. The preliminary analysis shows that the low soil water did affect radial stem growth but did not hinder photosynthesis - actually the highest GPP was recorded on that year, probably due to air temperature that was higher than usually. The presented model did not show decreased

performance during the low moisture conditions in 2013. Low temperatures with even freezing records visit the site almost every summer but those days do not pop up as decreased performance in the analysis either. These observations will be published in an independent manuscript (Matkala et al, under preparation)

What are the actions mentioned in the discussion (page 11, line 4)? Semantically action refers to purposeful and systematic interplay between components of the system! Although the action (or operation) of the system can be interpreted as evolutionary developed property of a living organism, the biochemical mechanism (enzymes, pigments, membrane pumps) as such lacks the purpose oriented action?

We consistently use the 'action' (of the biochemical regulation system) through the manuscript when discussing on synthesis or decomposition of the necessary, active compounds. There might be also some other term suitable such as 'functioning' but we are somewhat pleased with *action* since we believe that the tree actively regulates these substances which we can predict by the changes in the environmental factors.

In conclusion: the presented modelling is only a minor part of the research conducted during many years. The wider and more profound presentation of the study can be found in other printed sources. Material presented with current manuscript is an elegant demonstration of powerful methodological tools to create better comprehending of complex nature of living world. I do recommend to accept the paper with some modifications.

Minor comments I suggest some improvements to the abstract: the repetition of "theory" in concluding sentence should be avoided.

True, we reformulated it into the revised manuscript as "Our theory gained strong support in the rigorous test."

Number of measurements: 30000 datapoints during a summer (page 9, line 18): is it connected to total record 130000 (page 11, line 30)?

We had a mistake there since in the number should be 130 000. It is corrected in the revised manuscript.

Acronyms at the Acknowledgement part are understandable only for very few specialist: nevertheless the Google can provide more or less correct hints. Still, what is SMARI?

We agree that the acknowledged acronyms are quite unclear for most readers but at the same time, they are not that essential for them either. We corrected the misspelled SMARI to SMEAR I.

**A list of all relevant changes made in the manuscript**

- We clarified the scale of our interest (the daily and seasonal cycle in an individual branch) in the abstract and throughout the manuscript.
- We improved the wording and structure of several definitions and axioms.
- We improved the description of the needed measurements as well as the parameter estimation in the subchapter 2.3
- We included a discussion on the meaning of the parameters

**Annual cycle in Scots pine's photosynthesis**

Pertti Hari1, Veli-Matti Kerminen2, Liisa Kulmala1, Markku Kulmala2, Steffen Noe3, Tuukka Petäjä2, Anni Vanhatalo1, Jaana Bäck1

[revised manuscript text omitted]
     |               | Deleted: , and                                                            |
| 15 | the state of the photosynthetic machinery.                                                                                   | · · · · · · ( | Deleted: (i.e. enzymes, membrane pumps and pigments)                      |
|    | Further, we specify the relationship between environment and the synthesis by the biochemical regulation system.             | ·····(        | Deleted: annual                                                           |
|    | Axiom 2. The synthesis and activation of the photosynthetic machinery depend linearly on the temperature above freezing      |               | Deleted: enzymes, membrane pumps and pigments                      |
|    | point.                                                                                                                       | . (           |                                                                           |
|    | We clarify also the behaviour of decomposition and deactivation.                                                             |               |                                                                           |
| 20 | Axiom 3. The decomposition and deactivation of the photosynthetic machinery depends linearly on the state.                   |               | Deleted: enzymes, membrane pumps and pigments                             |
|    | Captured light energy may cause damage in chloroplasts in freezing temperatures, when availability of CO2 is limited for the |               |                                                                           |
|    | carbon reactions in photosynthesis. This is why the biochemical regulation system acts strongly to protect against damage.   |               |                                                                           |
|    | Axiom 4. The accelerated decomposition and deactivation of the photosynthetic machinery during cold and strong light         |               | Deleted: enzymes, membrane pumps and pigments                             |
|    | depends linearly on the product of light and temperature below freezing point.                                               |               |                                                                           |
| 25 | The concentrations of the photosynthetic machinery affect the performance of photosynthesis.                                 |               | Deleted: pigments, membrane pumps and enzymes                      |
|    |                                                                                                                              |               |                                                                           |

Definition 4. The efficiency of photosynthetic reactions is the capacity of light and carbon reactions to synthesise sugars.

When we developed the theory of photosynthesis explaining the behaviour in midsummer (Hari et al. 2014), we introduced an axiom stating that the product of saturating response to the photosynthetically active radiation and CO2 concentration in the stomatal cavity determines the photosynthesis at a point in space and time. Here, we introduce the annual cycle of photosynthesis into the axioms with the efficiency of photosynthetic carbon and light reactions and the efficiency entermediate product the average the average the product the formula (1) in the formula (1) and (2) and (2) and (3) and (3)

5 photosynthetic reactions replace the parameter b in the Eq.(1) in Hari et al. 2014.

**Axiom 5.** The photosynthesis rate at a point in space and time depends on the product of two terms: i) the efficiency of photosynthetic light and carbon reactions, and ii) the product of  $CO_2$  concentration in the stomatal cavity and the saturating response of the light reactions to the photosynthetically active radiation.

The state of the photosynthetic machinery determines the efficiency of photosynthetic light and carbon reactions, which leads to our final axiom:

Axiom 6. The efficiency of photosynthetic light and carbon reactions depends linearly on the state of the photosynthetic machinery.

**2.2. Mathematical analysis**

We introduce mathematical symbols to formulate exactly the axioms in a more exact and compact way. Let S denote the

15 state of the photosynthetic machinery,  $f_1$  is the synthesis and activation,  $f_2$  is the decomposition and deactivation,  $\underline{f_2}$  is the accelerated decomposition and deactivation of photosynthetic machinery (i.e. enzymes, membrane pumps and pigments) caused by light at low temperatures.

Axiom 2 states that the relationship between the synthesis and activation and temperature (T) is linear above the freezing point, which gives:

**20 $f_1(T) = Max(0, a_1(T + T_f))$**

where  $T_f$  is the freezing temperature of needles and  $a_i$  is a parameter.

According to axiom 3, the relationship between the decomposition and deactivation of photosynthetic machinery and the state of photosynthetic machinery, *S* is linear:

 $f_2(S) = a_2 S$

(2)

(1)

25 Accelerated decomposition and deactivation takes place to protect the photosynthetic machinery against damage when freezing temperatures hinders the carbon assimilation reactions of photosynthesis (Axiom 4): Deleted: of enzymes, membrane pumps and pigments

depends linearly

**$f_3(I,T) = a_3 \max\{(T_f - T) | I, 0\}$**

(3)

(6)

where I is the intensity of photosynthetically active radiation.

The synthesis, activation, decomposition and deactivation change the state of the photosynthetic machinery, as follows:

$$\frac{ds}{dt} = f_1 - f_2 - f_3 \tag{4}$$

5 Combining Equations (1)-(4), we obtain:

$$\frac{dS}{dt} = \max\{0, a_1(T + T_f)\} - a_2 S - a_3 \max\{(T_f - T) | I, 0\}$$
(5)

Equation (5) defines the state of the photosynthetic machinery at any moment t when temperature and solar radiation records are available.

The photosynthesis rate, p, is obtained from the axiom 5, as follows:

10  $p = E f(I) C_s$ ,

...

where  $C_s$  is the CO2 concentration in the stomatal cavity, f(I) is the saturating response of the photosynthesis rate to the photosynthetically active radiation (see Hari et al 2014), and *E* is the efficiency of photosynthetic carbon and light reactions which, according to the axiom 6, it is as follows:

| $E = a_4 S$ | (7) |
|-------------|-----|
|-------------|-----|

- 15 When we developed the theory of photosynthesis in midsummer (Hari et al. 2014), we introduced an axiom stating that the product of saturating response to the photosynthetically active radiation and CO2 concentration in the stomatal cavity determines the photosynthesis at a point in space and time (A1 in Hari et al. 2014). When we quantified the previous axiom with mathematical notations, we replaced the axiom A1 with the new axiom 5 that is quite similar with the previous one. The changing efficiency of photosynthetic light and carbon reactions is the novel aspect in the axiom 6. When we quantified 20 with mathematical notations the previous axiom, we introduced a parameter *b* (Eq. 1 in Hari et al. 2014). Equation (6) is very similar with the previous Eq. (1) in Hari et al. (2014); the only difference is that the efficiency parameter *b* is replaced
  - with E, the state variable efficiency of photosynthetic carbon and light reactions. We obtain the solution of the optimisation problem in the same way as in the analysis of photosynthesis (p) during midsummer, as follows:

 $p(\boldsymbol{l}, \boldsymbol{E}) = \frac{(u_{opt} g_{max} C_a + \boldsymbol{r}) a_4 S f(\boldsymbol{l})}{u_{opt} g_{max} + a_4 S f(\boldsymbol{l})}$

(8)

[revised manuscript text omitted]